# Aqueous Heat Method for the Preparation of Hybrid Lipid–Polymer Structures: From Preformulation Studies to Protein Delivery

**DOI:** 10.3390/biomedicines10061228

**Published:** 2022-05-24

**Authors:** Natassa Pippa, Nefeli Lagopati, Aleksander Forys, Maria Chountoulesi, Hektor Katifelis, Varvara Chrysostomou, Barbara Trzebicka, Maria Gazouli, Costas Demetzos, Stergios Pispas

**Affiliations:** 1Department of Pharmaceutical Technology, Faculty of Pharmacy, National and Kapodistrian University of Athens, Panepistimioupolis Zografou, 15771 Athens, Greece; mchountoules@pharm.uoa.gr (M.C.); demetzos@pharm.uoa.gr (C.D.); 2Department of Histology and Embryology, Medical School, National Kapodistrian University of Athens, 11527 Athens, Greece; nlagopati@med.uoa.gr; 3Centre of Polymer and Carbon Materials, Polish Academy of Sciences, 34 ul. M. Curie-Skłodowskiej, 41-819 Zabrze, Poland; aforys@cmpw-pan.edu.pl (A.F.); btrzebicka@cmpw-pan.edu.pl (B.T.); 4Laboratory of Biology, Department of Basic Medical Science, School of Medicine, National Kapodistrian University of Athens, 11527 Athens, Greece; e-ktor@hotmail.com (H.K.); mgazouli@med.uoa.gr (M.G.); 5Theoretical and Physical Chemistry Institute, National Hellenic Research Foundation, 48 Vassileos Constantinou Avenue, 11635 Athens, Greece; chrisostomou.varvara@gmail.com (V.C.); pispas@eie.gr (S.P.)

**Keywords:** cationic lipids, block copolymer, bovine serum albumin, differential scanning calorimetry (DSC), cryogenic transmission electron microscopy (cryo-TEM)

## Abstract

Liposomes with adjuvant properties are utilized to carry biomolecules, such as proteins, that are often sensitive to the stressful conditions of liposomal preparation processes. The aim of the present study is to use the aqueous heat method for the preparation of polymer-grafted hybrid liposomes without any additional technique for size reduction. Towards this scope, liposomes were prepared through the combination of two different lipids with adjuvant properties, namely dimethyldioctadecylammonium (DDA) and D-(+)-trehalose 6,6′-dibehenate (TDB) and the amphiphilic block copolymer poly(2-(dimethylamino)ethyl methacrylate)-b-poly(lauryl methacrylate) (PLMA-b-PDMAEMA). For comparison purposes, PAMAM dendrimer generation 4 (PAMAM G4) was also used. Preformulation studies were carried out by differential scanning calorimetry (DSC). The physicochemical characteristics of the prepared hybrid liposomes were evaluated by light scattering and their morphology was evaluated by cryo-TEM. Subsequently, in vitro nanotoxicity studies were performed. Protein-loading studies with bovine serum albumin were carried out to evaluate their encapsulation efficiency. According to the results, PDMAEMA-b-PLMA was successfully incorporated in the lipid bilayer, providing improved physicochemical and morphological characteristics and the ability to carry higher cargos of protein, compared to pure DDA:TDB liposomes, without affecting the biocompatibility profile. In conclusion, the aqueous heat method can be applied in polymer-grafted hybrid liposomes for protein delivery without further size-reduction processes.

## 1. Introduction

Dimethyldioctadecylammonium (DDA) is a cationic lipid with several applications in the field of nanomedicines and drug delivery systems [1,2,3,4,5,6,7,8,9]. DDA lipid nanoparticles can encapsulate self-amplifying RNA to be protected from degradation and to achieve efficient delivery and strong antigen-specific humoral and cellular-mediated immune responses to virtually any infectious disease [4]. DDA ligand-targeted formulations encapsulated plasmid DNA in the interior of a 100–150 nm pegylated liposome were also formed [5]. In other words, DDA cationic lipid is a raw material for the formulation of different types of systems in various structures and sizes for the delivery of nucleic acids and active pharmaceutical ingredients (APIs) of differing molecular weights [1,2,3,4,5]. The DDA-based systems can also be used as vaccine platforms with adjuvant properties [4].

Furthermore, an adjuvant system intended for mucosal vaccination based on biodegradable poly(DL-lactic-co-glycolic acid) (PLGA) nanoparticles modified with the DDA and the immunopotentiator trehalose-6,6′-dibehenate (TDB) lipids was designed and developed by a quality-by-design approach to tailor humoral and cellular immunity characterized by antibodies and Th1/Th17 responses [1].

Additionally, another interesting delivery platform category for the treatment of diseases is lipid-dendrimer hybrid nanosystems. Poly(amidoamine) (PAMAM) G4.0 with zwitterionic lipids were reported as delivery systems for paclitaxel with efficacy in killing ovarian cancer cells [10]. These systems are ideal for understanding in depth the different mechanisms involved in cellular processes, such as membrane fusion, transmembrane permeability, and endocytosis [11]. The different localizations of dendrimers in the cellular uptake have already been studied using the lipid bilayer models in vitro. The size and the surface charge correlated with the molecular structure are crucial parameters for dendrimer internalization and intracellular trafficking in living cells [12,13].

Apart from the formulation parameters and the properties of the different functional biomaterials that should be considered during the liposomal development, the preparation process also plays a key role. Conventional techniques, such as the aqueous heat method, are ideal for the easy scale up of the liposomal and lipidic nanoparticles for industrial production due to the absence of potentially toxic organic solvents, the elimination of the usually required two or three steps in the preparation protocol, and the high reproducibility of the processes [14,15,16]. The selection of lipids/excipients and of other materials, such as surfactants, are the critical quality parameters which impact the size and the polydispersity index of these lipid nanocarriers [17,18]. These physicochemical characteristics as well as the core-bilayer structure, the surface charge, the stability of the final formulation and the cargo encapsulation efficiency are also related to the formulation process and the quality aspects of the final product [14,15,16,17,18,19,20]. Therefore, in the present study, the aqueous heating method was used without any additional technique for size reduction to produce hybrid lipid-polymer nanostructures for protein encapsulation.

The aim of this investigation is to use the aqueous heat method for the preparation of polymer-grafted nanostructures without any additional technique for size reduction. The raw materials are: lipids with adjuvant properties: DDA and TDB; and amphiphilic block copolymer poly(2-(dimethylamino)ethyl methacrylate)-b-poly(lauryl methacrylate) (PLMA-b-PDMAEMA). This amphiphilic block copolymer has been already used for the formulation of stimuli-responsive lyotropic liquid crystalline nanosystems and stimuli-responsive polymer-grafted liposomes for the delivery of hydrophobic anti-glioma agents [21,22,23]. For comparison reasons, we also used PAMAM G4. PAMAM G4 is a dendritic polymer containing amine groups with a different internal chemical structure and topology compared to the linear PLMA-b-PDMAEMA block copolymer (Figure 1). On the other hand, PAMAM dendrimers are commercially available, and many studies have already been published regarding their interactions with lipid bilayers and the preparation of lipid-dendrimer nanostructures for drug loading and controlled release. Therefore, it is of interest to study dendrimers as polymeric components of the formulations using the aqueous heat method. DSC was used for the preformulation studies to identify the interactions between lipids/polymers and lipid/dendrimers. Light-scattering techniques and cryogenic transmission electron microscopy (cryo-TEM) were used for the evaluation of the prepared systems. Nanotoxicity and protein-loading studies were designed as a road map for the evaluation of the prepared systems as carriers for delivery purposes.

## 2. Materials and Methods

### 2.1. Materials

Dimethyldioctadecylammonium (DDA) (Figure 1a) and D-(+)-trehalose 6,6′-dibehenate (TDB) (Figure 1b) were purchased from Avanti Polar Lipids Inc. (Albaster, AL, USA); chloroform and methanol was purchased from LabScan (Dublin, Ireland). The PDMAEMA-b-PLMA (Figure 1c) amphiphilic diblock copolymer was synthesized with RAFT polymerization methodologies [23]. PAMAM dendrimer generation 4 (PAMAM G4) (Figure 1d) (empirical formula C_622_H_1248_N_250_O_124_, formula weight: 14,214.17) was purchased from Aldrich. Bovine serum albumin was purchased from Merck (Darmstadt, Germany) and Pierce™ (Pittsburgh, PA, USA). The BCA protein assay kit was purchased from Thermo Scientific™, Waltham, MA, USA.

### 2.2. Methods

#### 2.2.1. Differential Scanning Calorimetry

DSC experiments were performed on an 822e Mettler-Toledo (Schwerzenbach, Switzerland) calorimeter calibrated with pure indium (Tm = 156.6 °C) and water. Sealed aluminum 40 μL crucibles were used as sample holders. DDA:TDB 1:0.2; DDA:TDB:PLMA-b-PDMAEMA (1:0.2:0.5; 1:0.2:1; and 1:0.2:2.5 weight ratios) and DDA:TDB:PAMAM G4 (1:0.2:0.1) bilayers were dissolved in chloroform:methanol 9/1 *v/v*, and the solvents were removed by slow evaporation. Residual solvents were removed under vacuum overnight. The dried material was weighted into the aluminum crucibles, and lipid bilayers were prepared by hydration using water for injections. An empty aluminum crucible was used as reference. Prior to measurements, the crucibles were heated at a temperature that exceeds the transition of DPPC (41.7 °C) to ensure equilibration. Three heating-cooling cycles were performed: 10° to 60 °C at 5 °C/min or 20 °C/min scanning rate, respectively. A thermodynamic evaluation was performed during the second cooling and heating cycle. All samples were scanned until identical curves were obtained. Errors were evaluated based on at least three replicate runs. Enthalpy changes and characteristic transition temperature were calculated using Mettler-Toledo STARe software.

#### 2.2.2. Preparation of Lipid Nanostructures; Polymer–Lipid Nanostructures; and Lipid-Dendrimer Nanostructures

All the nanostructures were prepared using the aqueous heat method [7].

#### 2.2.3. Cryogenic Transmission Electron Microscopy (cryo-TEM) Measurements

Cryogenic transmission electron microscopy (cryo-TEM) images were obtained using a Tecnai F20 X TWIN microscope (FEI Company, Hillsboro, OR, USA) equipped with a field emission gun, operating at an acceleration voltage of 200 kV. Images were recorded on the Gatan Rio 16 CMOS 4k camera (Gatan Inc., Pleasanton, CA, USA) and processed with Gatan Microscopy Suite (GMS) software (Gatan Inc., Pleasanton, CA, USA). The specimen preparation was done by vitrification of the aqueous solutions on grids with holey carbon film (Quantifoil R 2/2; Quantifoil Micro Tools GmbH, Großlöbichau, Germany). Prior to use, the grids were activated for 15 s in oxygen plasma using a Femto plasma cleaner (Diener Electronic, Ebhausen, Germany). Cryo-samples were prepared by applying a droplet (3 μL) of the suspension to the grid, blotting with filter paper, and immediately freezing in liquid ethane using a fully automated blotting device Vitrobot Mark IV (Thermo Fisher Scientific, Waltham, MA, USA). After preparation, the vitrified specimens were kept under liquid nitrogen until they were inserted into a cryo-TEM-holder Gatan 626 (Gatan Inc., Pleasanton, CA, USA) and analyzed in the TEM at −178 °C.

#### 2.2.4. In Vitro Cytotoxicity

HEK293 cells were cultivated in DMEM high glucose culture medium (BioSera), containing 10% FBS, 2 mmol/L glutamine, 100 U/mL penicillin, and 100 g/mL streptomycin at 37 °C. Changes of medium were performed every 48 h, and cells were passaged on a weekly basis using the standard trypsin EDTA method. When cells reached a sufficient confluency, they were transferred to a 96-well plate (corning-Costar, Corning, NY, USA) with approximately 5000 cells/well. The MTS assay was used (CellTitre 96R Aqueous MTS, Promega) to quantify the reduction of viability following exposure to prepared nanosystems. The principle of this assay is based on the action of mitochondrial dehydrogenase enzymes that consume NAD(P)H to reduce a tetrazolium compound (MTS) to formazan, and the concentration of the latter is proportional to the number of living cells. Its concentration is calculated via absorbance reading at 490 nm. Each of the nanosystems were tested in seven different concentrations (from 25 to 500 μg/mL) and during an 16 h incubation. Additionally, three types of controls were used: a positive control (cells that were not exposed to nanostructures), a negative control (nanosystems in cell-free medium), and a background control (culture medium only). Absorbances were normalized with respect to the untreated control cultures to calculate changes in cell viability. All experiments were held in duplicate to ensure reproducibility.

#### 2.2.5. BSA Encapsulation and Encapsulation Efficiency (%EE) Studies

The encapsulation protocol for the loading of all the prepared systems with BSA is described in [24]. Briefly, DDA:TDB 1:0.2 liposomes; DDA:TDB:PLMA-b-PDMAEMA (1:0.2:0.5; 1:0.2:1, and 1:0.2:2.5 weight ratios) polymer–lipid nanostructures and DDA:TDB:PAMAM G4 lipid dendrimer nanostructures (1:0.2:0.1 weight ratio) were mixed with BSA for 30 min. The colloidal concentration of the prepared systems was 2 mg/mL, and the concentration of BSA was 1 mg/mL. After the electrostatic complexation process, we studied the physicochemical characteristics of the complexes formed using dynamic and electrophoretic light scattering (DLS, ELS).

The free BSA was separated from the BSA entrapped in the nanostructures by using the ultrafiltration centrifugal method. In detail, the nanostructures were centrifuged for 30 min at 4000 rpm using centrifugal filter tubes [molecular weight (MW) cutoff = 100 kDa; Millipore]. The particles were separated from the aqueous phase, and the free BSA was analyzed in the supernatant by the bicinchoninic acid (BCA) protein quantification method according to the kit protocol.

Protein quantification using BCA (Pierce™ BCA Protein Assay Kit, Thermo Scientific™) protein assay was carried out following manufacturer’s instructions. Briefly, samples were incubated up to 30 min at 37 °C, with 25 µL of sample and 200 µL of the working reagent. Absorbance was then measured at 562 nm using an INNO microplate reader.

Centrifuged samples of the respective empty nanostructures were also conducted by BCA assay and used as a blank. The loading % was calculated according to the following equation:$$\left(loading\right)\;\%=\left(1-\frac{C_{supernatant}}{C_{total}}\right)\%$$
where *C_supernatant_* is the BSA that was quantified in the supernatant (non-entrapped), and *C_total_* is the total concentration of the BSA added in the dispersion [25].

## 3. Results and Discussion

### 3.1. Preformulation Studies: The Role of the Guest into DDA:TDB Lipid Bilayer Thermotropic Behavior

The first step for the design and the development of hybrid/chimeric nanostructures composed by different materials is to investigate the interactions of the guest molecule with the lipid bilayer. DSC is widely used for qualification and quantification of these interactions and is a useful tool for the later formulation studies. In Table 1, the calorimetric values obtained during the heating and the cooling processes are presented. According to the literature, the DSC scan for pure DDA is characterized by one sharp well-defined endothermic peak with main transition temperature at 47.2 °C in 10 mM Tris at pH 7.4 [7,19,20]. The incorporation of TDB in different molar ratios causes shift in T_m_ values toward lower temperatures and have been already studied in the literature [7]. We should pointed out that the addition of the polymeric guest caused alterations of the thermotropic characteristics of the DDA:TDB, but the curves did not show significant changes. The curves are more or less the same as those published in [7]. In our study, we run DSC experiments of lipid bilayers hydrated in HPLC-grade water. For DDA:TDB lipid bilayers, we observed the T_m_ at 51.4 °C and the pre-transition (T_s_) at 40.3 °C. These small differences in main and pre-transition temperatures are related to the difference in the dispersion medium. In previous studies, Tris buffer was used, and in this investigation the DSC experiments were run in HPLC-grade water. The presence of salts caused different hydration networks on the surface of lipid bilayers and consequently a slight difference in the T_m_ values. The presence of PLMA-b-PDMAEMA caused a decrease of the T_m_ values around 10 °C for all the weight ratios studied. The decrease of T_s_ was around 7 °C for the 1:0.2:0.5 and 1:0.2:1 weight ratios. At the highest weight ratio of the polymeric guest (i.e., 1:0.2:2.5), the abolishment of the pretransition phenomenon was observed. The enthalpy increased slightly (in absolute values) with the incorporation of polymeric chains into DDA:TDB lipid bilayers (from −117.0 J/mol for DDA:TDB to −113.7 J/mol for DDA:TDB:PLMA-b-PDAMAEMA at a 1:0.2:0.5 weight ratio) (Table 1). The increase of the PLMA-b-PDAMAEMA weight ratio caused a significant reduction (in absolute values) of the ΔH values (for DDA: TDB:PLMA-b-PDAMAEMA 1:0.2:1 ΔH = −46.5 J/mol and for 1:0.2:2.5 ΔH = −42.6 J/mol) (Table 1). The significant decrease of the T_m_ and ΔH values implies that PLMA hydrophobic polymer chains are incorporated into the lipid bilayer, disrupting its structure by creating new hydrophobic interactions. The cooperativity of the systems is significant as ΔΤ_1/2_ values also decreased (Table 1). The decrease or the abolishment of the pretransition leads to the conclusion that the hydrophilic polymeric chain also interacts with the polar groups of the lipids, causing a different orientation of the polar head due to the creation of hydrophilic interactions between PDMAEMA and DDA/TDB. In other words, the presence of the PLMA-b-PDMAEMA copolymer controls the molecular packing of the lipids by altering their physical liquid crystalline phase, dynamics, and phase separation [26,27,28].

During the cooling of the systems, the DDA:TDB lipid bilayer and the DDA:TDB:PLMA-b-PDMAEMA showed a significant hysteresis in the cooling process, as indicated by the T_m_ values (Table 1). This phenomenon of the non-thermodynamic reversibility during the heating and cooling processes is well established in the literature [29]. The ΔH values were decreased in the presence of the polymeric guest into lipid bilayers. The reduction was higher as the weight ratio of the polymer increased (Table 1). According to our previous published data, the crystallization of the polymer/lipid membrane takes place in lower temperatures due to restrictions to the mobility of the lipids caused by the presence of the block copolymer [30].

We also investigated the impact of PAMAM G4 in the thermotropic behavior of DDA:TDB lipid bilayers. The presence of the dendrimer caused significant reductions in the T_m_ and the ΔH values. We should underline that the ΔH is close to zero (Table 1). These results showed that the incorporation of PAMAM G4 caused disruption of the organization of the lipid bilayer. Additionally, the ΔΤ_1/2_ value increased in comparison to all the other studied systems, and this implies that there is no cooperativity between the lipids and the dendrimer. The same results were obtained from the cooling process (Table 1). According to the DSC data, the formation of DDA:TDB-coated PAMAM G4 complexes is achieved due to the strong interactions of the positively charged dendrimer and the TDB sugar head groups (absence of the pretransition) and the hydrophobic interactions between the lipid chains of the DDA:TDB and the arms of PAMAM G4 (reduction of Tm and ΔH values) [12,13].

### 3.2. Formulation Studies: The Role of the Guest on DDA:TDB Liposome Size and Morphology

We studied by DSC how the PLMA-b-PDMAEMA and the PAMAM G4 alter the thermotropic behavior of DDA:TDB lipid membranes. The outcomes of the preformulation studies were a road map for the design and the development of nanostructures composed of the same materials. We used the aqueous heat method for the preparation of pure and mixed nanostructures. To the best of the authors’ knowledge, this is the first report in the literature that this protocol is utilized for the preparation of polymer–lipid nanostructures and dendrimer–lipid nanostructures. We used dynamic and electrophoretic light scattering to investigate the alterations of the physicochemical characteristics of the pure liposomes in the presence of the polymer or dendrimer guest.

In Figure 2, the size, the ζ-potential, and the scattering intensity of the prepared nanostructures are shown. The size distribution graphs are included in Appendix A. The pure DDA:TDB were about 600 nm with high polydispersity and extremely positive ζ-potential (around 40 mV). These values are more or less in accordance with the previously published data. The pure lipid systems were visualized by cryo-TEM images as spherical “solid” objects, sometimes joined together (Figure 3a). Their diameter was found around 50 nm with cryo-TEM. This difference is probably related to the inability for observing the hydration layer in the cryo-TEM measurements of the liposome characteristics and the presence of the aggregation phenomena in the dispersion state. The tendency of the liposomes to become joined together is the reason that we observed higher values of D_h_ by DLS measurements. This liposomal system was prepared for comparison reasons. The incorporation of the polymeric guest at different weight ratios caused a significant reduction of the size of the prepared nanostructures (Figure 2a). As the weight ratio of PLMA-b-PDMAEMA block copolymer increased, the size decreased. At the highest weight ratio, polymer–lipid nanostructures with sizes below 200 nm were observed. The population of the particles became more homogenous with the increasing weight ratio of PLMA-b-PDMAEMA (Appendix A). The ζ-potential values increased but not significantly (Figure 2b), probably due to the partially protonated amino groups of PDMAEMA block at the pH 5.5 of HPLC-grade water [21]. The scattering-intensity value of DDA:TDB:PLMA-b-PDMAEMA 1:0.2:0.5 increased significantly in comparison to the scattering-intensity value of the pure liposomes (from ≈80 KCps to ≈140 KCps). This observation is evidence of the successful incorporation of the copolymer into the liposomal membrane. In other words, the scattering-intensity value depends on many parameters and also on composition; thus, the changes reflect the particle sizes and particle composition. As the sizes decreased, we can conclude the changes of particle composition were caused by incorporation of the copolymer. This observation is in agreement with the DSC results during the preformulation studies, where the significant decrease of the T_m_ and ΔH values for 1:0.2:0.5 ratio implied that PLMA hydrophobic polymer chains are incorporated into the lipid bilayer. The increase of the weight ratio of PLMA-b-PDMAEMA did not lead to a significant increase of the scattering-intensity values. On the other hand, we observed a small decrease (Figure 2c). In our opinion, there may be a certain percentage of polymers which are incorporated in the lipid membrane with which the saturation of the lipid bilayer with polymer loading is achieved. The amount of polymer that is not incorporated can form other structures, which are in a smaller percentage and do not contribute significantly to the increase in the mass of the system, which is expressed through the intensity values. Alternatively, the size decrease of the polymer–lipid nanostructures or changes in their structure may also result in a decrease of the overall scattering intensity of the system.

The cryo-TEM imaging showed that the incorporation of the polymer caused the formation of different hybrid structures. These structures and their sizes were strongly dependent on the weight ratio of the PLMA-b-PDMAEMA copolymer (Figure 3b–d). Namely, for DDA:TDB:PLMA-b-PDMAEMA 1:0.2:0.5 weight ratio, vesicles with spherical and irregular shapes with sizes between 70–270 nm and irregular shaped particles with strong contrast, joined on the surfaces of the vesicles were observed (Figure 3b). For DDA:TDB:PLMA-b-PDMAEMA 1:0.2:0.5 weight ratio, the vesicles were between 70–290 nm (in accordance to light-scattering measurements) and “disc” like particles were also observed (Figure 3c). At the highest weight ratio of the polymeric guest, vesicles covered with “threadlike” structures with sizes of 50–260 nm were observed (Figure 3d). It may be that the aforementioned amount of polymer that is not fully incorporated in the highest ratio, as indicated by light-scattering measurements can contribute to these “threadlike” structures that cover the vesicles. After all, the total disappearance of the pretransition found in the highest ratio 1:0.2:0.5 in the DSC measurements suggests the creation of hydrophilic interactions between PDMAEMA and DDA/TDB that are more intensive than those at the lowest ratios, where the pretransition was decreased but still present. The incorporation of amphiphilic block copolymers to lipid bilayers leads from sphere to ellipsoid deformation and disc formation due to different molecular packing of the lipids [24,27,31]. In other words, it is true that the structures containing the polymer are deformed. It is rather difficult to see purely ellipsoidal structures and disks. For all the polymer–lipid nanostructures, we should point out that the thickness of the membrane was 5–6 nm (Figure 3). The thickness of the membrane was determined by the analysis of cryo-TEM images. The observed thickness is in accordance with the recent published data from other groups and our previous studies [32,33,34]. Moreover, the tendency of the pure liposomes to be joined together, yielding a D_h_ and PDI increase, was prevented by the incorporation of the PDMAEMA-b-PLMA copolymer. Incorporation of the copolymer may cause steric repulsion due to the formation of a hydrophilic PDMAEMA polymeric corona, as it is depicted by the gradual size decrease and the presence of individual particles, as the polymer ratio in the system is increased.

In Figure 4, the cryo-TEM images of DDA:TDB:PAMAM G4 are presented. The hybrid system self-assembled in different structures. Namely, spherical (Figure 4a) and irregular (Figure 4b) shape vesicles were formed with sizes of 20–60 nm and 60–450 nm, respectively. The thickness of the membrane is 5–7 nm for the lipid/copolymer hybrid discussed above and is greater than the thickness of pure liposomes. Large aggregates are also observed (Figure 4c). The cryo-TEM images are in accordance with high polydispersity values that were obtained from the DLS measurements and visualized from the size distribution graphs (Appendix A). Lipid-PAMAM G4 hybrid nanoparticles with spherical morphology have been presented in the literature. On the other hand, according to the translocation mechanism, PAMAM dendrimers can open holes in lipid bilayers by stealing lipids from the bilayer and forming “dendrisomes” [12,13]. This mechanism is the driving force for the formation of large aggregates with sizes around 1000 nm, according to cryo-TEM and DLS measurements. According to Mecke et al. 2004, the dendrimers sometimes remove lipid molecules from the substrate and form aggregates consisting of a dendrimer surrounded by lipid molecules [35]. The structures around 1000 nm are probably composed of PAMAM G4 dendrimers with DDA:TDB lipids all around them. This is another possible mechanism for the formation of holes in the lipid bilayer [35].

We also tested the stability of the prepared systems for a period of three months. As mentioned before, the final dispersions were placed in a glass vial and stored in the refrigerator at 4 °C. Appendix A shows the results of the stability assessment experiments using DLS. Pure DDA:TDB liposomes were stable for one week. After this period, an increase of the particle size was observed. Sedimentation phenomena were observed 40 days after the preparation of the nanosystems. For this reason, in Appendix A, data until 30 days are presented. For DDA:TDB:PLMA-b-PDMAEMA 1:0.2:0.5 and 1:0.2:1 weight ratios, the sizes remained unaffected for at least a month (Appendix A). After this period, a small increase in the particle size was observed, but no sedimentation, nor aggregation phenomena took place. The system with the highest weight ratio remained stable for three months. The size and the size distribution of the DDA:TDB:PLMA-b-PDMAEMA 1:0.2:2.5 weight ratio nanosystem did not change (Appendix A). The last observation means that the presence of the polymeric guest and its weight ratio are of paramount importance for the long-term stability of the systems. In our opinion, the steric repulsion of the hydrophilic polymeric blocks is responsible for the long-term stability of the dispersions because all these systems exhibited high ζ-potential values (Figure 2b), a characteristic which is absent in the pure liposomes case. For this reason, pure liposomes were stable only for one week. Furthermore, for the DDA:TBD:PAMAM G4 systems, aggregation phenomena took place two days after the preparation. These aggregates were observed in the cryo-TEM images. Sedimentation phenomena took place seven days after preparation. It may be concluded that the dendritic macromolecule does not infer stability to the hybrid nanosystems as the linear amphiphilic PLMA-b-PDMAEMA copolymer does.

### 3.3. Nanotoxicity Studies and Protein Loading Properties

The nanotoxicity profiles of all the prepared systems were tested. In Figure 5, the cell viability vs. different concentrations of the prepared systems at different weight ratios is presented. Based on the MTS assay, all five samples showed a dose-dependent toxicity on the HEK293 cell lines. The lowest toxicity levels were exhibited by sample DDA:TDB:PLMA-b-PDMAEMA (1:0.2:2.5) since even the maximum concentration tested (500 μg/mL) resulted in a viability that exceeded 80% (Figure 5). The incorporation of the block copolymer at the highest weight ratio exhibited the higher biocompatibility results in the HEK293 cell lines. On the contrary, DDA:TDB:PAMAM G4 had the most cytotoxic effects. Even from the lowest concentration (25 μg/mL), its toxicity was higher than that of all other samples (approximately 70%), while the highest concentration was cytotoxic for half of the cellular population exposed. According to Liu et al., 2015, the majority of the lipids in the lipid–dendrimer structures are in the plasma membrane, while the PAMAM dendrimers can be distributed intracellularly upon uptake. This mechanism could be a possible explanation for the nanotoxicity profile of DDA:TDB: PAMAM G4 systems. The cytotoxic profile of all the other systems are rather similar with the pure DDA:TDB to be the least cytotoxic. We should also point out that the nanotoxicity of pure DDA:TDB carriers is higher than other liposomes prepared with zwitterionic lipids, i.e., DPPC, DSPC, etc. [36]. For this reason, the incorporation of the PLMA-b-PDMAEMA in the lipid bilayer led to a system with more or less the same biocompatibility profile at the concentrations tested. Thus, the incorporation of the PDMAEMA-b-PLMA successfully improved the physicochemical and morphological characteristics of the liposomes to more homogenous dispersions of smaller sizes and organized morphologies but without burdening the biocompatibility profile, which all are very promising properties for future pharmaceutical applications. The fact that these results were obtained using the aqueous heating method and by just simply increasing the copolymer content with no need for any additional technique for size reduction reinforces the above statement.

Considering, nanotoxicity studies of the prepared systems, we tested their ability to load protein molecules. We used BSA as a model protein/antigen. We kept the colloidal concentration of the pure carriers at 2 mg/mL and the concentration of the protein utilized at 1 mg/mL. The physicochemical characteristics of the systems with BSA are presented in Table 2, and the size distribution of the complexes are illustrated in Appendix A. The pure liposomes did not load BSA. We observed aggregates with sizes around 10 mμ, and the scattering intensity was decreased dramatically. This phenomenon means that the presence of BSA destroyed the vesicular/liposomal structure of the pure DDA:TDB systems. On the other hand, the presence of PLMA-b-PDMAEMA block copolymer stabilized the lipid bilayer, and polymer-grafted lipoplexes were obtained. The weight ratio of PLMA-b-PDMAEMA copolymer played a key role on the physicochemical characteristics of the resulted structures. Namely, the size of the BSA complexes decreased as the weight ratio of the PLMA-b-PDMAEMA increased (Table 2). In all cases, the scattering intensity increased significantly (more than 100 KCps) in the presence of protein, showing successful attachment of the BSA on the surface of the polymer–lipid nanostructures. Additionally, another crucial parameter showing the successful complexation of BSA onto the polymer–lipid nanostructure surface is the ζ-potential (Table 2, Figure 2). The ζ-potential values of the complexes decreased around 30 mV in the presence of the model protein. This is expected due to electrostatic interactions between the cationic polymer–lipid nanostructures and the anionic protein surface. The size distribution became also more homogeneous at the highest weight ratio of the PLMA-b-PDMAEMA (Appendix A). The encapsulation efficiency of the BSA increased with the increase of the weight ratio of PLMA-b-PDMAEMA copolymer. The increase of positively charged PDMAEMA probably blocks on the surface of hybrid lipid-polymer structures at the higher polymer ratios and enhances the electrostatic interactions between the cationic polymer–lipid nanostructures and the anionic protein surface. In other words, considering all the above results, the role of the polymeric guest is crucial for the utilization of these systems for protein-delivery purposes. The resulting complexes also have the ideal size and size distribution for intramuscular and/or nasal administration.

For the DDA:TBD:PAMAM G4 systems, we observed increased size after BSA complexation, a small reduction of the ζ-potential values, reduction of the mass (I values decreased), heterogeneous size distribution, and the highest loading efficiency (Table 2, Appendix A). The size of the DDA:TDB:PAMAM G4:BSA was around 2700 nm, and the EE is 93%. The different surface characteristics due to the presence of the dendrimer (DSC studies and cryo-TEM images) and the formation of supramolecular aggregates between the lipid–dendrimer–BSA complexes are probably the reason for the obtained physicochemical characteristics. BSA can form complexes with dendrimer–lipid surfaces via electrostatic interactions and after that aggregates or flocculates can be formed. For this reason, sizes of the complexes around 2700 nm with high polydispersity were observed.

## 4. Conclusions

In this study, we used the aqueous heat method without size reduction protocol but in low-energy, non-harsh, non-organic solvent conditions for the development of hybrid polymer–lipid delivery platforms with enhanced physicochemical and structural characteristics useful for protein delivery. To the best of our knowledge, this is the first report in the literature where this preparation protocol is used as the formulation process of nanostructures of the type with detailed knowledge of their physicochemical, morphological, thermotropic, nanotoxicity, and protein-loading characteristics.

## Figures and Tables

**Figure 1 biomedicines-10-01228-f001:**
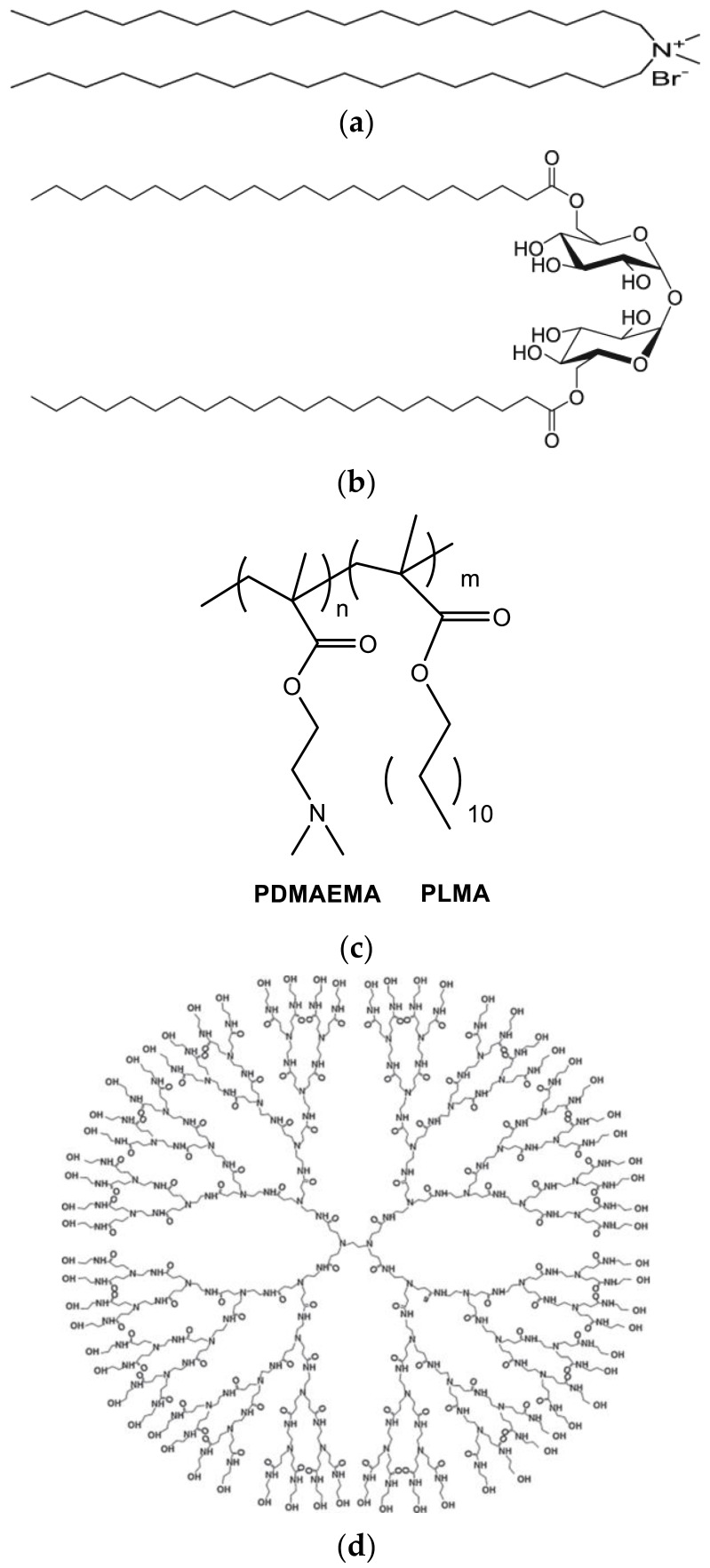
The chemical structures of (**a**) DDA, (**b**) TDB, (**c**) PLMA-b-PDMAEMA, and (**d**) PAMAM-G4.

**Figure 2 biomedicines-10-01228-f002:**
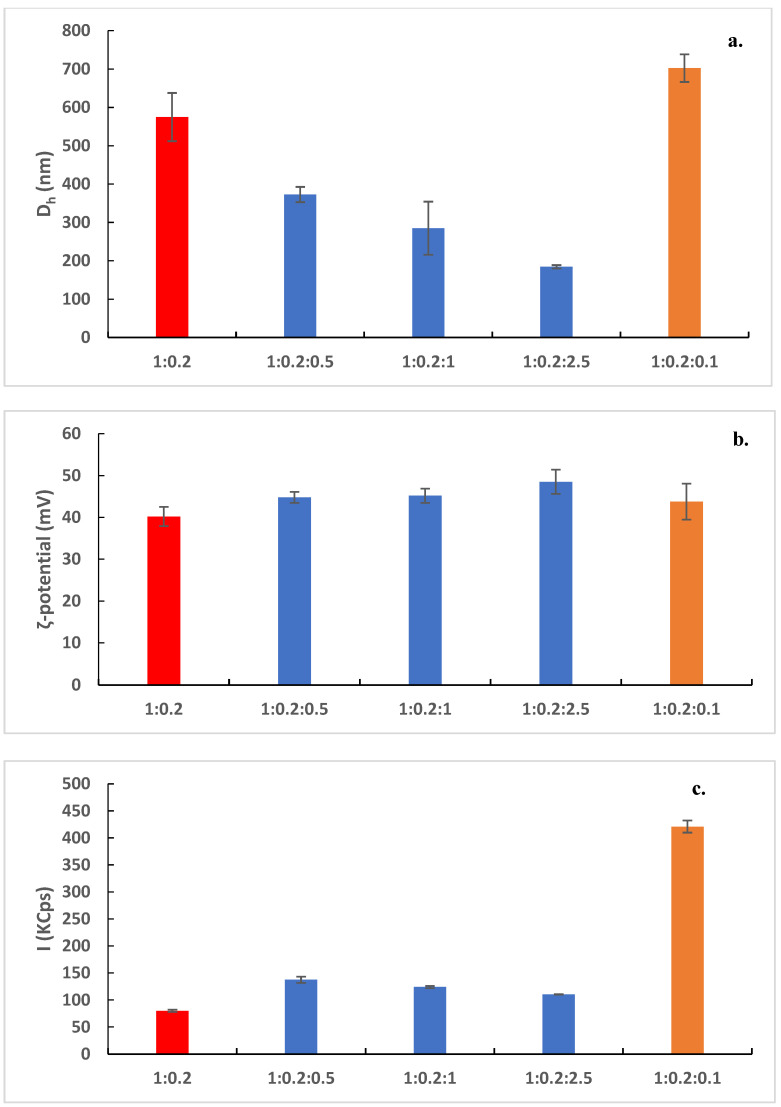
(**a**) Hydrodynamic diameter, D_h_ (nm), (**b**) ζ-potential (mV), and (**c**) scattering intensity I (KCps) of DDA:TDB 1:0.2 (red bars); DDA:TDB:PLMA-b-PDMAEMA (1:0.2:0.5; 1:0.2:1 and 1:0.2:2.5 weight ratios) (blue bars); and DDA:TDB:PAMAM G4 (1:0.2:0.1 weight ratio) nanostructures (orange bars).

**Figure 3 biomedicines-10-01228-f003:**
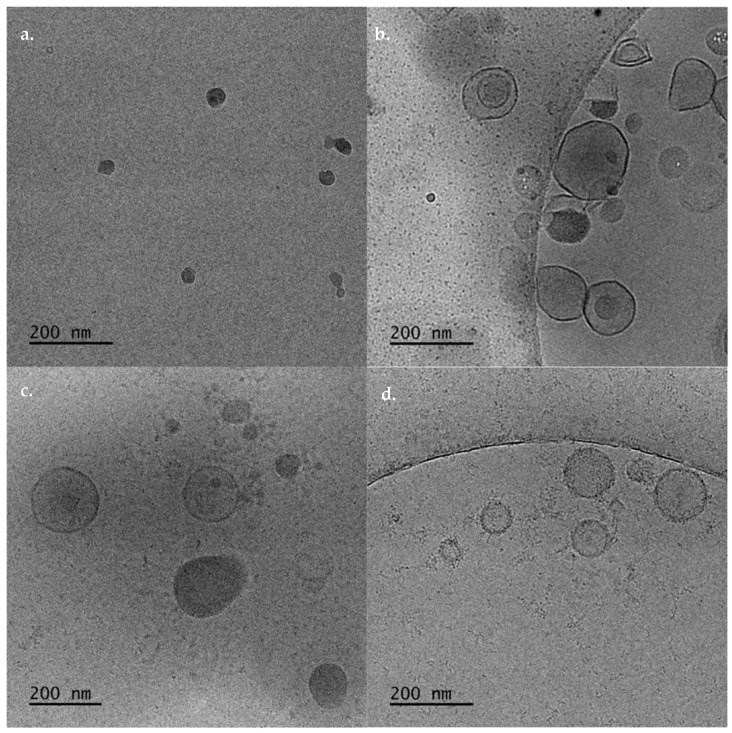
Cryo-TEM images of (**a**) DDA:TDB, DDA:TDB:PLMA-b-PDMAEMA polymer lipid nanostructures at (**b**) 1: 0.2:0.5, (**c**) 1:0.2:1, and (**d**) 1:0.2:2.5 weight ratios.

**Figure 4 biomedicines-10-01228-f004:**
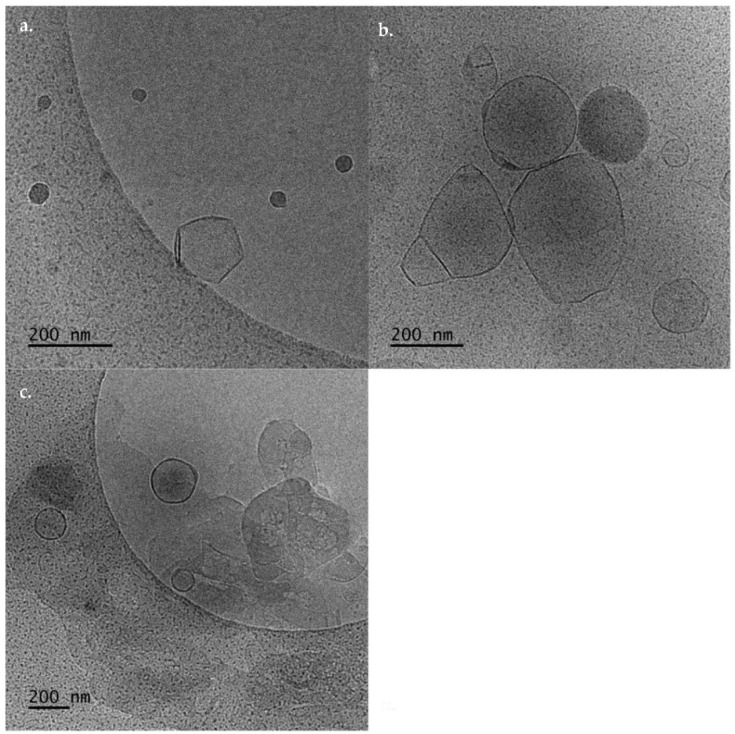
Cryo-TEM images of DDA:TDB:PAMAM G4 showing structures of (**a**) spherical shape solid structures, (**b**) spherical and irregular vesicular shapes, and (**c**) aggregates.

**Figure 5 biomedicines-10-01228-f005:**
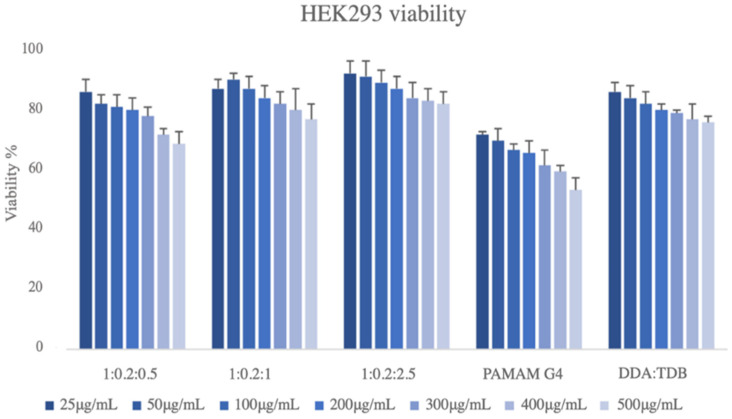
Cell viability vs. different concentrations of the prepared hybrid systems of DDA:TDB 1:0.2; DDA:TDB:PLMA-b-PDMAEMA (1:0.2:0.5; 1:0.2:1, and 1:0.2:2.5 weight ratios) and DDA:TDB:PAMAM G4 (1:0.2:0.1 weight ratio) nanostructures.

**Table 1 biomedicines-10-01228-t001:** Calorimetric values of lipid bilayers.

Sample	Weight Ratio	Tonset.m (°C) ^a^	T_m_ (°C) ^b^	ΔΤ1/2.m (°C) ^c^	ΔH_m_ (J/mol) ^d^	Tonset.s (°C)	T_s_ (°C)	ΔΤ1/2 (°C)	ΔH_s_ (KJ/mol)
**Heating**									
DDA:TDB	1:0.2	49.7	51.4	1.69	−117.0	38.5	40.3	1.71	−109
DDA:TDB:PLMA-b-PDMAEMA	1:0.2:0.5	36.8	39.5	2.98	−133.7	32.6	33.8	2.54	−274
DDA:TDB:PLMA-b-PDMAEMA	1:0.2:1	38.8	40.3	1.43	−46.5	29.2	30.9	1.97	−19
DDA:TDB:PLMA-b-PDMAEMA	1:0.2:2.5	37.6	40.3	2.06	−42.6	-	-	-	-
DDA:TDB:PAMAM G4	1:0.2:0.1	26.6	37.9	16.19	−4.5	-	-	-	-
**Cooling**									
DDA:TDB	1:0.2	37.7	37.6	1.68	363.9	-	-	-	-
DDA:TDB:PLMA-b-PDMAEMA	1:0.2:0.5	40.4	37.4	5.84	613.9	-	-	-	-
DDA:TDB:PLMA-b-PDMAEMA	1:0.2:1	40.0	37.4	4.57	409.3	-	-	-	-
DDA:TDB:PLMA-b-PDMAEMA	1:0.2:2.5	42.8	35.5	7.01	447.2	-	-	-	-
DDA:TDB:PAMAM G4	1:0.2:0.1	43.5	38.2	8.0	84.8	-	-	-	-

^a^ T_onset_: temperature at which the thermal event starts; ^b^ T.: temperature at which heat capacity (ΔC_p_) at constant pressure is maximum; ^c^ ΔT_1/2_: half width at half height of the transition peak; ^d^ ΔH: transition enthalpy normalized per mol of lipid bilayer system. m: main transition; s: secondary transition.

**Table 2 biomedicines-10-01228-t002:** The physicochemical characteristics of the complexes with BSA.

System	Weight Ratio	D_h_ (nm)	Ζ-Potential (mV)	I (KCps)	% Loading
DDA:TDB:PLMA-b-PDMAEMA:BSA	1:0.2:0.5:1	418.8 ± 29.5	14.8 ± 1.6	203.4 ± 1.7	82.0 ± 3.9
DDA:TDB:PLMA-b-PDMAEMA:BSA	1:0.2:1:1	277.9 ± 13.9	13.2 ± 8.1	240.8 ± 1.9	88.7 ± 5.7
DDA:TDB:PLMA-b-PDMAEMA:BSA	1:0.2:2.5:1	231 ± 25	18.2 ± 1.7	221.7 ± 1.7	91.5 ± 1.8
DDA:TDB:PAMAM G4:BSA	1:0.2:0.1:1	2721.1 ± 153.6	32.3 ± 6.2	130.7 ± 3.7	93.0 ± 8.9

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
