# Peer review of "Aqueous Heat Method for the Preparation of Hybrid Lipid–Polymer Structures: From Preformulation Studies to Protein Delivery"

_biomedicines, 2022, doi:10.3390/biomedicines10061228_

Round 1

Reviewer 1 Report

The authors tried to revise the manuscript according to the comments of the reviewers. Again, it seems that the authors conducted various experiments, but the results are descriptive and VERY redundant and it is hard to read. I think the length of the total manuscript should be extensively shortened to about 2/3 or less. Please consider whether to include everything what you did in one manuscript to make the conclusion statement. All the sections, Introduction, Materials and Methods, and Results and Discussion are too long to read. Moreover, the section of Conclusion can be shortened more. The Conclusion starts with the aim of this study and it seems like an Abstract. I have never seen such a long conclusion. The entire manuscript should be revised to make it concise.

The design of Figure 1 should be considered. The resolution the dendrimer is poor and it seem the figure of dendrimer is a copy. On the other hand, I wonder the description and data of dendrimer is necessary or not because the particle size in Table 2 is terrible and obviously not successful.

Reviewer 2 Report

The manuscript entitled "Aqueous Heat Method for the Preparation of Lipid-Polymer Hybrid Structures: From Preformulation Studies to Protein Delivery" has serious flaws.
When a copolymer is used in combination with lipids, hybrid vesicles can form, the term liposome cannot be used in this case.
The encapsulation that the authors are talking about is in fact a covering of the vesicles with BSA.

Reviewer 3 Report

The manuscript is based on the protein delivery by liposomes. Wherein, polymer has been employed to make the hybrid liposomes. An interesting pre-formulation study was done to choose the best lipid bilayer composition. The manuscript is well organised and covers all required aspects. Therefore, it can be accepted for the publication.

However, one query should be fulfilled concerning the encapsulation efficiency.

In the table 2, the average values for the %EE should be presented along with the standard deviation.

Good luck!

Round 2

Reviewer 1 Report

The authors tried to shorten the entire manuscript including Conclusion. Please consider the following minor points to be revised.

1. Figure 1 is comprised of four molecules, and the style of the chemical structure should be unified at least for a, b, and c. 

2. Figure 1 is separated into two pages. This can be combined into one page.

3. In Figure 2a, the Y-axis label, Dh --> Dh

4. In Figure 5, the Y-axis label shows "Viability %", then erase "%" for all the values, such as 100% --> 100.

5. In the legend of Figure 5 please indicate the meanings of the composition ratios, such as 1:0.2:0.5, as indicated in other Figure legends.

Reviewer 2 Report

The authors did not take into account the received comments.

In the text, the name "liposomes" is still present when "hybrid liposomes" should be used.

"Encapsulation" should be replaced with "loading".

Author Response

This manuscript is a resubmission of an earlier submission. The following is a list of the peer review reports and author responses from that submission.

Round 1

Reviewer 1 Report

In this work, the authors prepared liposomes with  dimethyldioctadecylammonium (DDA), D-(+)-trehalose 6,6'-dibehenate (TDB) and the amphiphilic block copolymer poly(2-(dimethylamino)ethyl methacrylate)-b-poly(lauryl methacrylate) (PLMA-b-PDMAEMA). The liposomes were characterized by differential scanning calorimetry (DSC), light scattering and cryo-TEM. Overall, this work is lack of novelty, I do not recommend this work to publish in this journal.

  1. The authors should revise the introduction section and reconsider the logical relationship between the introduction and the work.
  2. If the authors want to emphasize the aqueous heat method, more experiment and description is needed. Otherwise, the title should be carefully considered.
  3. Considering the PAMAM is a dendritic polymer, both the chemical structure and topology are very different from the PLMA-b-PDMAEMA, the authors should explain why PAMAM is selected to be compared.
  4. The authors claimed the thickness of the membrane was 5-6 nm. It is important to explain how the authors get this data.
  5. If the authors want to apply these liposomes to practical applications, more experiments like stability, cargo release are needed,

Reviewer 2 Report

In this paper is reported liposome preparations from various combinations of synthetic amphiphiles including DDA, copolymers, and dendrimers. The liposomes are simply prepared by heating and shaking by hand. The physicochemical characterization was performed by conventional methods. Protein encapsulation efficiency was evaluated by using stable albumin solution.

It seems that the authors conducted various experiments, but the results are descriptive and redundant. What is the main product as a result? The entire paper could be more concise. The conclusion section (line 463-498) is too long.

  1. Title

In this study, liposomes are prepared by simply heating and shaking. Therefore, the results may not be reproducible, because no info is given how strongly it was shaken. Magnet stirring at a certain RPM, constant solution volume and stirring time would be more reproducible.  “Aqueous Heat Method” does not seem special and may not be necessary to be emphasized in the title.

  1. Abstract

The authors conclude “the aqueous heat method can be applied in polymer grafted hybrid liposomes for protein delivery without further reduction processes”. However, it is well known that a simple lipid dispersion can produce liposomes, and the particle sizes listed in Table 2 are larger than 200 nm and cannot be filtrated for sterilization. Some other processes would be needed for size reduction. What is the main conclusion of this study??

  1. Page 5, line 194

Provide reference paper of Luo et al., 2017.

  1. BCA Protein assay kit, line 215-217

To measure Ctotal of the liposomes containing BSA, the turbidity of liposomal solution should interfere with the results. Liposomes have to be destroyed in advance.

  1. DSC data

The authors summarized the data of DSC in Table 1. However, the graphs would be more understandable for readers who are not familiar with the profile of DDA.

  1. Line 277-280

“The formation of DDA:TDB coated PAMAM G4 complexes is achieved due to the strong interactions of positively charged dendrimer and the polar head group…”  Both dendrimer and DDA are positively charged and they should be repulsive with each other.

6. Figure 2 is too large in size and can be minimized by shortening the distances between the bars, for example.